# Discouraging posterior collapse in hierarchical Variational Autoencoders using context

## Abstract

Hierarchical Variational Autoencoders (VAEs) are among the most popular likelihood-based generative models. There is a consensus that the top-down hierarchical VAEs allow effective learning of deep latent structures and avoid problems like posterior collapse. Here, we show that this is not necessarily the case, and the problem of collapsing posteriors remains. To discourage this issue, we propose a deep hierarchical VAE with a context on top. Specifically, we use a Discrete Cosine Transform to obtain the last latent variable. In a series of experiments, we observe that the proposed modification allows us to achieve better utilization of the latent space and does not harm the model's generative abilities.

## 1 Introduction

Latent variable models (LVMs) parameterized with neural networks constitute a large group in deep generative modeling (Tomczak, 2022). One class of LVMs, Variational Autoencoders (VAEs) (Kingma & Welling, 2014; Rezende et al., 2014), utilize amortized variational inference to efficiently learn distributions over various data modalities, e.g., images (Kingma & Welling, 2014), audio (Van Den Oord et al., 2017) or molecules (Gómez-Bombarelli et al., 2018). One of the problems hindering the performance of VAEs is the *posterior collapse* (Wang et al., 2021) when the variational posterior (partially) matches the prior distribution (e.g., the standard Gaussian distribution). The expressive power of VAEs could be improved by introducing a hierarchy of latent variables. The resulting hierarchical VAEs like ResNET VAEs (Kingma et al., 2016), BIVA (Maaløe et al., 2019), very deep VAE (VDVAE) (Child, 2021) or NVAE (Vahdat & Kautz, 2020) achieve state-of-the-art performance on images in terms of the negative log-likelihood (NLL). Despite their successes, hierarchical VAEs could still suffer from the posterior collapse effect. As a result, the modeling capacity is lower, and some latent variables carry very little to no information about observed data.

In this paper, we take a closer look into the posterior collapse in the context of hierarchical VAEs. It was claimed that introducing a specific *top-down* architecture of variational posteriors (Sønderby et al., 2016; Maaløe et al., 2019; Child, 2021; Vahdat & Kautz, 2020) solves the problem and allows learning powerful VAEs. However, we can still notice at least partial posterior collapse, where some of the latent variables are completely ignored by the model. Here, we fill a few missing gaps in comprehending this behavior. We analyze the connection between posterior collapse and latent variable non-identifiability. By understanding the issue that lies in the optimization nature of the Kullback-Leibler terms, we propose to utilize a non-trainable, discrete, and deterministic transformation (e.g., Discrete Cosine Transform) to obtain informative top-level latent variables. Making the top latent variables highly dependent on data, we alter the optimization process. The resulting hierarchical VAE starts utilizing the latent variables differently. In the experiments, we show that our proposition achieves a different landscape of latent space.

The contributions of the paper are the following:

- We provide empirical evidence that the posterior collapse is present in top-down hierarchical VAEs (Section 3.2).
- We extend the analysis of the posterior collapse phenomenon presented by (Wang et al., 2021) to hierarchical VAEs (Section 3.3).
- We propose a way to discourage posterior collapse by introducing Discrete Cosine Transform (DCT) as a part of the variational posterior (Section 4).

- In the experiments, we show that the proposed approach leads to better latent space utilization (Section 5.2), more informative latent variables (Section 5.3) and does not harm the generative performance (Section 5.1).

## 2 BACKGROUND

### 2.1 VARIATIONAL AUTOENCODERS

Consider random variables $\mathbf{x} \in \mathcal{X}^D$ (e.g., $\mathcal{X} = \mathbb{R}$). We observe $N$ $\mathbf{x}$'s sampled from the empirical distribution $q(\mathbf{x})$. We assume that each $\mathbf{x}$ has $L$ corresponding latent variables $\mathbf{z}_{1:L} = (\mathbf{z}_1, \ldots, \mathbf{z}_L), \mathbf{z}_l \in \mathbb{R}^{M_l}$, where $M_l$ is the dimensionality of each variable. We aim to find a latent variable generative model with unknown parameters $\theta$, $p_\theta(\mathbf{x}, \mathbf{z}_{1:L}) = p_\theta(\mathbf{x}|\mathbf{z}_{1:L})p_\theta(\mathbf{z}_{1:L})$. In general, optimizing latent-variable models with non-linear stochastic dependencies is troublesome. A possible solution is an approximate inference in the form of variational inference (Jordan et al., 1999) with a family of variational posteriors over the latent variables $\{q_\phi(\mathbf{z}_{1:L}|\mathbf{x})\}_\phi$. This idea is exploited in Variational Auto-Encoders (VAEs) (Kingma & Welling, 2014; Rezende et al., 2014), in which variational posteriors are referred to as encoders. As a result, we optimize a tractable objective function, i.e., the Evidence Lower BOund (ELBO), over the parameters of the variational posterior, $\phi$, and a generative part, $\theta$:

$$\mathbb{E}_{q(\mathbf{x})}\left[\ln p_\theta(\mathbf{x})\right] \geq \mathbb{E}_{q(\mathbf{x})}\left[\mathbb{E}_{q_\phi(\mathbf{z}_{1:L}|\mathbf{x})}\ln p_\theta(\mathbf{x}|\mathbf{z}_{1:L}) - D_{\mathrm{KL}}\left[q_\phi(\mathbf{z}_{1:L}|\mathbf{x})\|p_\theta(\mathbf{z}_{1:L})\right]\right], \quad (1)$$

where $q(\mathbf{x})$ is an empirical data distribution. Further, we use $q^{\text{test}}(\mathbf{x})$ for the hold-out data.

### 2.2 TOP-DOWN HIERARCHICAL VAEs

We propose to factorize the distribution over the latent variables in an autoregressive manner: $p_\theta(\mathbf{z}_1, \ldots, \mathbf{z}_L) = p_\theta(\mathbf{z}_L)\prod_{l=1}^{L-1} p_\theta(\mathbf{z}_l|\mathbf{z}_{l+1:L})$, similarly to (Child, 2021; Maaløe et al., 2019; Vahdat & Kautz, 2020). Next, we follow the proposition of (Sønderby et al., 2016) with the top-down inference model: $q_\phi(\mathbf{z}_1, \ldots, \mathbf{z}_L|\mathbf{x}) = q_\phi(\mathbf{z}_L|\mathbf{x})\prod_{l=1}^{L-1} q_\phi(\mathbf{z}_l|\mathbf{z}_{l+1:L}, \mathbf{x})$. This factorization was used previously by successful VAEs, among others, NVAE (Vahdat & Kautz, 2020) and Very Deep VAE (VDVAE) (Child, 2021). It was shown empirically that such a formulation allows for achieving state-of-the-art performance on several image datasets.

## 3 AN ANALYSIS OF THE *posterior collapse* IN HIERARCHICAL VAEs

The *posterior collapse* effect is a known problem of shallow VAEs when certain latent variables do not carry any information about the observed data. There are various methods to deal with this issue for VAEs, such as changing the parameterization (Dieng et al., 2019; He et al., 2019), changing the optimization or the objective (Alemi et al., 2018; Bowman et al., 2016; Fu et al., 2019; Havrylov & Titov, 2020; Razavi et al., 2019), or using hierarchical models (Child, 2021; Maaløe et al., 2017; 2019; Tomczak & Welling, 2018; Vahdat & Kautz, 2020). Here, we focus entirely on the hierarchical VAEs since the posterior collapse problem is not fully analyzed in their context.

In practice, hierarchical VAEs usually require huge latent space with multiple latent layers to achieve good performance (Sønderby et al., 2016; Maaløe et al., 2019; Child, 2021; Vahdat & Kautz, 2020). However, as we show in our analysis, the actual number of used latent units in these models is relatively small. Therefore, it is still an open question about how to reduce the gap between the total size of the latent space and the actual number of latents used by these models.

Following definition 1 in Wang et al. (2021), we consider the posterior collapse as a situation where the true posterior is equal to the prior for a given set of parameters $\theta$. We can formulate this definition for a single stochastic layer of top-down hierarchical VAE as follows:

$$p_\theta(\mathbf{z}_l|\mathbf{z}_{l+1:L}, \mathbf{x}) = p_\theta(\mathbf{z}_l|\mathbf{z}_{l+1:L}). \quad (2)$$

In practice, we deal with the variational posterior $q_\phi(\mathbf{z}_l|\mathbf{z}_{l+1:L}, \mathbf{x})$, which approximates the true posterior. Furthermore, it is common to identify the posterior collapse based on this approximate distribution (Burda et al., 2015; Lucas et al., 2019; Sønderby et al., 2016; Van Den Oord

Table 1: Posterior collapse metrics and NLL for the top-down hierarchical VAEs with various LATENT SPACE sizes and with fixed model SIZE (the total number of parameters).

| SIZE | L | LATENT SPACE | AU | KL | NLL↓ |
|------|---|--------------|------|-------|------|
| 676K | 4 | 490 | 38.3% | 0.047 | 79.6 |
| 624K | 6 | 735 | 37.9% | 0.031 | 78.8 |
| 657K | 8 | 980 | 33.5% | 0.022 | 78.3 |
| 651K | 10 | 1225 | 33.6% | 0.018 | 77.9 |

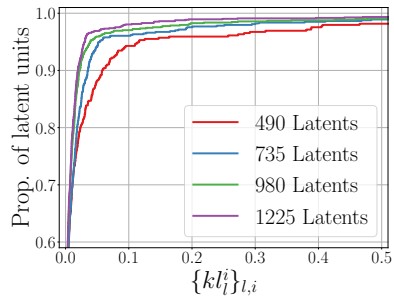

Figure 1: The cumulative distribution function of the KL-divergence in VAEs with varying latent space sizes.

et al., 2017). Both definitions are connected, yet not identical. We learn the posterior approximation by variational inference, and the ELBO (Eq. 1) is maximized when the approximate posterior matches the true posterior, namely, $D_{\mathrm{KL}}\left[q_\phi(\mathbf{z}_{1:L}|\mathbf{x})\|p_\theta(\mathbf{z}_{1:L}|\mathbf{x})\right] = 0$. Furthermore, the KL-divergence can be further decomposed into the following sum: $D_{\mathrm{KL}}[q_\phi(\mathbf{z}_{1:L}|\mathbf{x})\|p_\theta(\mathbf{z}_{1:L}|\mathbf{x})] = D_{\mathrm{KL}}\left[q_\phi(\mathbf{z}_L|\mathbf{x})\|p_\theta(\mathbf{z}_L|\mathbf{x})\right] + \sum_{l=1}^{L-1}\mathbb{E}_{q_\phi(\mathbf{z}_{l+1:L},\mathbf{x})}D_{\mathrm{KL}}\left[q_\phi(\mathbf{z}_l|\mathbf{z}_{l+1:L},\mathbf{x})\|p_\theta(\mathbf{z}_l|\mathbf{z}_{l+1:L},\mathbf{x})\right].$

Therefore, a collapsed true posterior distribution for the latent variable at the stochastic layer $l$ results in a collapsed variational posterior for this latent variable at the optimum. However, the collapse of the variational posterior distribution does not guarantee the collapse of the true posterior as it can be caused by a poor choice of the family of the variational distributions. See Appendix A for an in-depth discussion. To this end, we assume that the family of variational posterior distribution is rich enough and use the variational posterior collapse as an indicator of true posterior collapse. Next, we discuss the metrics of the posterior collapse in more detail.

## 3.1 MEASURING THE POSTERIOR COLLAPSE

We consider two metrics for assessing the posterior collapse in hierarchical VAEs. First, we compute the *KL-divergence* for the $i$-th latent variable of the stochastic layer $l$:

$$\mathrm{kl}_l^i = \mathbb{E}_{q^{\mathrm{test}}(\mathbf{x})}\mathbb{E}_{q_\phi(\mathbf{z}_{l+1:L}|\mathbf{x})}D_{\mathrm{KL}}\left[q_\phi(\mathbf{z}_l^i|\mathbf{z}_{l+1:L},\mathbf{x})\|p_\theta(\mathbf{z}_l^i|\mathbf{z}_{l+1:L})\right]. \tag{3}$$

This quantity can be approximately computed using Monte Carlo sampling and gives us an estimate of the posterior collapse issue for each latent variable. Note that the KL-divergence term used in the ELBO 1 equals the sum of these values over all latent variables $i$ and stochastic layers $l$.

Second, we use *active units*. This is a metric introduced in (Burda et al., 2015), and it can be calculated for a given stochastic layer and a threshold $\delta$:

$$\mathrm{A}_l = \mathrm{Var}_{q^{\mathrm{test}}(\mathbf{x})}\mathbb{E}_{q_\phi(\mathbf{z}_{l+1:L}|\mathbf{x})}\mathbb{E}_{q_\phi(\mathbf{z}_l|\mathbf{z}_{l+1:L},\mathbf{x})}\left[\mathbf{z}_l\right], \tag{4}$$

$$\mathrm{AU} = \frac{\sum_{l=1}^{L}\sum_{i=1}^{M_l}\left[\mathrm{A}_{l,i} > \delta\right]}{\sum_{l=1}^{L}M_l}, \tag{5}$$

where $M_l$ is the dimensionality of the stochastic layer $l$, $[P]$ is Iverson bracket, which equals to 1 if $P$ is true and to 0 otherwise. Following (Burda et al., 2015), we use the threshold $\delta = 0.01$. The higher the share of active units, the more efficient the model is in using its latent space.

## 3.2 EMPIRICAL EVIDENCE OF POSTERIOR COLLAPSE

In the following, we carry out an experiment to observe the posterior collapse in hierarchical VAE. We train four top-down hierarchical VAE models with different latent space sizes on the MNIST dataset. At the same time, we make sure that all the models have a similar number of parameters and try to keep the number of ResNet blocks the same. We vary the number of stochastic layers $L$ from 4 to 10. Note that the data space has a dimensionality of 784. We report the test NLL, Active Unit, and KL-divergence per latent variable for this experiment in Table 1. We also plot an empirical CDF of the latent variable's KLs in Figure 1.

The total number of latent units increases from 490 to 1225 in this experiment. However, all the models have no more than 40% of active units. We also observe that AU and KL metrics decrease with the number of stochastic layers increasing. The cumulative histogram of KL-divergence (Eq. 3) depicted in Figure 1 shows that the models have close to 60% of the latent variable with almost zero KL-divergence. This indicates that the deep hierarchical VAEs do not use the majority of the latent units. As a result, the common claim that the top-down hierarchical VAEs alleviate the problem of the posterior collapse (Maaløe et al., 2019) is not necessarily true as indicated by this experiment. It is true, though, that increasing the number of latents improves the performance (NLL). However, this is not an efficient way of utilizing the model since it disregards over 60% of its latents.

### 3.3 LATENT VARIABLES NON-IDENTIFIABILITY AND THE POSTERIOR COLLAPSE IN HIERARCHICAL VAEs

Wang et al. (2021) prove that collapse of the true posterior in a one-level VAE takes place if and only if latent variables are *non-identifiable*. A latent variable $\mathbf{z}$ is called non-identifiable (Raue et al., 2009) if for a given set of parameter values $\theta^*$, the conditional likelihood does not depend on this latent variable. Namely, $p_{\theta^*}(\mathbf{x}|\mathbf{z}) = p_{\theta^*}(\mathbf{x})$. Similarly, we say that latent variable $\mathbf{z}_l$ in hierarchical VAE is non-identifiable when $p_{\theta^*}(\mathbf{x}|\mathbf{z}_{1:L}) = p_{\theta^*}(\mathbf{x}|\mathbf{z}_{-l})$.

We now establish the connection between posterior collapse (Eq. 2) and non-identifiability in the following propositions. See Appendix B for the proofs.

**Proposition 1** *Consider a top-down hierarchical VAE introduced in Section 2.2. Then, for a given set of parameter values $\theta^*$, the posterior of the latent variable $\mathbf{z}_l$ collapses if and only if $\mathbf{x}$ and $\mathbf{z}_l$ are conditionally independent given $(\mathbf{z}_{l+1}, \dots, \mathbf{z}_L)$.*

**Proposition 2** *Consider a top-down hierarchical VAE introduced in Section 2.2. If $\mathbf{x}$ and $\mathbf{z}_l$ are conditionally independent given $(\mathbf{z}_{l+1}, \dots, \mathbf{z}_L)$, then the latent variable $\mathbf{z}_l$ is non-identifiable. However, if $\mathbf{z}_l$ is non-identifiable, it does not imply that it is conditionally independent with $\mathbf{x}$ given $(\mathbf{z}_{l+1}, \dots, \mathbf{z}_L)$.*

To simplify the notation, let us split the latent variables of hierarchical VAEs into three groups:

$$\underbrace{\mathbf{z}_1, \dots, \mathbf{z}_{l-1}}_{\mathbf{z}_A}, \mathbf{z}_l, \underbrace{\mathbf{z}_{l+1}, \dots, \mathbf{z}_L}_{\mathbf{z}_C}. \tag{6}$$

We can do this for each $l \in 1, \dots, L$, assuming that in the corner case of $l = 1$, $\mathbf{z}_A$ is an empty set, and in the case of $l = L$, $\mathbf{z}_C$ is an empty set. Then, the content of the propositions 1 and 2 can be summarized in the following diagram:

$$\underbrace{p_{\theta^*}(\mathbf{z}_l|\mathbf{z}_C, \mathbf{x}) = p_{\theta^*}(\mathbf{z}_l|\mathbf{z}_C)}_{\text{Posterior Collapse}} \Leftrightarrow \underbrace{p_{\theta^*}(\mathbf{x}|\mathbf{z}_l, \mathbf{z}_C) = p_{\theta^*}(\mathbf{x}|\mathbf{z}_C)}_{\text{Conditional Independence}} \Rightarrow \underbrace{p_{\theta^*}(\mathbf{x}|\mathbf{z}_A, \mathbf{z}, \mathbf{z}_C) = p_{\theta^*}(\mathbf{x}|\mathbf{z}_A, \mathbf{z}_C)}_{\text{Non-identifiability}}.$$

That being said, as opposed to the one-level VAE considered by (Wang et al., 2021), the non-identifiability of the latent variables in hierarchical VAEs does not necessarily cause the true posterior to collapse. Therefore, the solution, in which we define the likelihood function in a way that guarantees the latent variable identifiability might be too restrictive. One possible solution would be to utilize the method from (Wang et al., 2021) to ensure that $\mathbf{z}_l$ and $\mathbf{x}$ are not conditionally independent given $(\mathbf{z}_{l+1}, \dots, \mathbf{z}_L)$. However, one would need access to the distribution $p_{\theta^*}(\mathbf{x}|\mathbf{z}_l, \mathbf{z}_{l+1:L})$, which is intractable in the top-down hierarchical VAEs.

As a result, we employ an orthogonal approach by adding one more *non-trainable* latent variable to a hierarchical VAE, which we call a *context*. We show in Section 4.3 that this method can break the link between conditional independence and posterior collapse without any restriction on the likelihood function.

## 4 HIERARCHICAL VAEs WITH NON-TRAINABLE CONTEXT

### 4.1 HIERARCHICAL VAEs WITH CONTEXT

In this work, we introduce a modified hierarchical VAE model, which is meant to increase the number of latent variables used by a deep hierarchical VAE while not harming performance. As

we discuss in Sec. 3.3, posterior collapse happens if and only if there is a conditional independence between $\mathbf{z}_l$ and $\mathbf{x}$ given $\mathbf{z}_{>l}$. If this is the case, then the posterior distribution is proportional to the prior, namely, $p_\theta(\mathbf{z}_l|\mathbf{z}_{l+1:L}, \mathbf{x}) \propto p_\theta(\mathbf{z}_l|\mathbf{z}_{l+1:L})$. As a result, the latent variable $\mathbf{z}_l$ does not contain any information about the input $\mathbf{x}$. Note also that prior distribution is an object we can control since this is the distribution we parametrize directly by the neural network. This motivates us to introduce the context. We think of the context as a top-level latent variable that can be obtained from the input via a fixed, non-trainable transformation.

Let us consider the top latent variable $\mathbf{z}_L$ to be given by a non-learnable transformation of the input $\mathbf{x}$, namely, $\mathbf{z}_L = f(\mathbf{x})$. We require context $\mathbf{z}_L$ to be a much simpler object than the initial object $\mathbf{x}$. That is, we want the dimensionality of $\mathbf{z}_L \in \mathbb{R}^{M_L}$ to be smaller than the dimensionality of $\mathbf{x} \in \mathcal{X}^D$, $M_L \ll D$. At the same time, we want the context to be a reasonable representation of $\mathbf{x}$. We can think of the context as a *compressed* representation of the input data, e.g., in the simplest case, it could be a downsampled version of an image (see Appendix F for details). We discuss another way of constructing the context in Section 4.4.

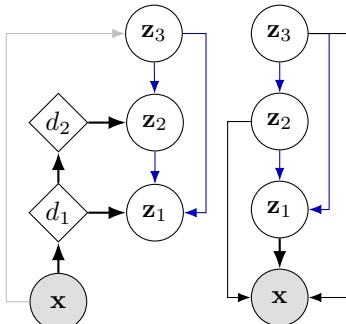

Figure 2: Graphical model of the top-down hierarchical VAE with two latent variables and the context $\mathbf{z}_3$. The inference model (left) and the generative model (right) share the top-down path (blue). The grey arrow represents a non-trainable transformation.

The graphical model of the VAE with the context is depicted in Figure 2. We use the top-down VDVAE architecture (Child, 2021) and extend this model with a deterministic, non-trainable function to create latent variable $\mathbf{z}_L$ (the context). Context $\mathbf{z}_L$ is produced from the observation $\mathbf{x}$ and further used to condition all other latent variables in both inference and generative models. We provide a mode details on the architecture in Appendix E.1 (Figure 8)

## 4.2 TRAINING VAE WITH THE CONTEXT

We assume that both $\mathbf{x}$ and $\mathbf{z}_L$ are discrete random variables. Furthermore, we assume that the variational posterior of the context is Kronecker's delta function $q(\mathbf{z}_L|\mathbf{x}) = \delta(\mathbf{z}_L - f(\mathbf{x}))$. As we depict in Figure 2, the generative model is conditioned on the context latent variable $\mathbf{z}_L$ at each step. To sample unconditionally, we define a context prior distribution $p_\gamma(\mathbf{z}_L)$, which is trained simultaneously with the whole VAE model via the ELBO objective. Following (Vahdat et al., 2021; Wehenkel & Louppe, 2021), we propose to use a diffusion-based generative model (Ho et al., 2020) as the prior. Since the context is a less complex object, we assume that it is enough to use a model much smaller compared to the VAE itself. We provide details on diffusion models in Appendix C. The diffusion-based model provides a lower bound on the log density of the prior distribution $\mathcal{L}(\gamma, \mathbf{z}_L) \leq \ln p_\gamma(\mathbf{z}_L)$, which together with VAE objective 1 results in the following objective:

$$\mathbb{E}_{q_\phi(\mathbf{z}_{1:L}|\mathbf{x})}\left[\ln p_\theta(\mathbf{x}|\mathbf{z}_{1:L})\right] + \mathbb{E}_{q(\mathbf{z}_L|\mathbf{x})}\mathcal{L}(\gamma, \mathbf{z}_L) - \sum_{l=1}^{L-1}\mathbb{E}_{q_\phi(\mathbf{z}_{l+1:L}|\mathbf{x})}D_{\mathrm{KL}}\left[q_\phi(\mathbf{z}_l|\mathbf{z}_{l+1:L}, \mathbf{x})\|p_\theta(\mathbf{z}_l|\mathbf{z}_{l+1:L})\right].$$

## 4.3 THE POSTERIOR COLLAPSE FOR VAES WITH THE CONTEXT

We claim that the introduction of the context changes the prior distributions, which results in the posterior collapse having less effect on the model. First, since $\mathbf{z}_L = f(\mathbf{x})$, we guarantee that the top latent variable will not collapse. We now need to fit the prior to the aggregated posterior $q(\mathbf{z}_L) = \sum_{\mathbf{x}} \delta(\mathbf{z}_L - f(\mathbf{x}))q(\mathbf{x})$, not the other way around. As a result, this prior contains information about the data points $\mathbf{x}$ by definition. Second, let us assume that $\mathbf{z}_l$ and $\mathbf{x}$ are conditionally independent for given parameter values $\theta^*$: $p_\theta(\mathbf{x}|\mathbf{z}_l, \mathbf{z}_{l+1:L}) = p_\theta(\mathbf{x}|\mathbf{z}_{l+1:L})$. Then, from the Proposition 1 the posterior is proportional to the prior: $p_\theta(\mathbf{z}_l|\mathbf{z}_{l+1:L}, \mathbf{x}) \propto p_\theta(\mathbf{z}_l|\mathbf{z}_{l+1:L})$. However, since $f(\mathbf{x}) = \mathbf{z}_L \in \mathbf{z}_{l+1:L}$, we still have information about $\mathbf{x}$ preserved in the posterior:

$$p_\theta(\mathbf{z}_l|\mathbf{z}_{l+1:L}, \mathbf{x}) \propto p_\theta(\mathbf{z}_l|\mathbf{z}_{l+1:L-1}, f(\mathbf{x})). \tag{7}$$

This way, the presence of posterior collapse does not necessarily lead to uninformative latent codes.

### 4.4 A DCT-BASED CONTEXT

We suggest to think of the context as of *compressed* representation of the input data (Sec. 4.1). We expect it to be lower-dimensional compared to the data itself while preserving crucial information. In other words, we may say that context does not contain any high-frequency details of the signal of interest while preserving a more general pattern. To this end, we propose to use the Discrete Cosine Transform[1] (DCT) to create the context. DCT(Ahmed et al., 1974) is widely used in signal processing for image, video, and audio data, i.e., it is a part of the JPEG standard (Pennebaker & Mitchell, 1992). DCT is a linear transformation that decomposes a discrete signal on a basis consisting of cosine functions of different frequencies.

Let us consider a signal as a $3D$ tensor $\mathbf{x} \in \mathcal{X}^{\text{Ch} \times D \times D}$. Then DCT for a single channel, $\mathbf{x}_i$, is defined as follows: $\mathbf{z}_{DCT,i} = \mathbf{C}\mathbf{x}_i\mathbf{C}^\top$, where for all pairs $(k = 0, n)$: $\mathbf{C}_{k,n} = \sqrt{\frac{1}{D}}$, and for all pairs $(k, n)$ such that $k > 0$: $\mathbf{C}_{k,n} = \sqrt{\frac{2}{D}} \cos\left(\frac{\pi}{D}\left(n + \frac{1}{2}\right)k\right)$. A helpful property of the DCT is that it is an *invertible* transformation. Therefore, it contains all the information about the input. However, for our approach, we want the context to be lower-dimensional compared to the input dimensionality. Therefore, we propose to remove high-frequency components from the signal. Assume that each channel of $\mathbf{x}$ is $D \times D$. We select the desired size of the context $d < D$ and remove (crop) $D - d$ bottom rows and right-most columns for each channel in the frequency domain. Finally, we perform normalization using matrix $\mathbf{S}$, which contains the maximal absolute value of each frequency. We calculate this matrix using all the training data: $\mathbf{S} = \max_{\mathbf{x} \in \mathcal{D}_{\text{train}}} |\text{DCT}(\mathbf{x})|$. As a result, we get latent variables whose values are in $[-1, 1]$. In the last step, we round all values to a given precision such that after multiplying the latents by $\mathbf{S}$ we get integers, thus, we get discrete variables. We call this the *quantization* step. Algorithm 1 describes context computation from the given input $\mathbf{x}$.

---

**Algorithm 1** Create a DCT-based context

**Input**: $\mathbf{x}, \mathbf{S}, d$
$\quad \mathbf{z}_{\text{DCT}} = \text{DCT}(\mathbf{x})$
$\quad \mathbf{z}_{\text{DCT}} = \text{Crop}(\mathbf{z}_{\text{DCT}}, d)$
$\quad \mathbf{z}_{\text{DCT}} = \frac{\mathbf{z}_{\text{DCT}}}{\mathbf{S}}$
$\quad \mathbf{z}_{\text{DCT}} = \text{quantize}(\mathbf{z}_{\text{DCT}})$
**Return**: $\mathbf{z}_{\text{DCT}}$

---

**Algorithm 2** Decode the DCT-based context.

**Input**: $\mathbf{z}_{\text{DCT}}, \mathbf{S}, D$
$\quad \mathbf{z}_{\text{DCT}} = \mathbf{z}_{\text{DCT}} \cdot \mathbf{S}$
$\quad \mathbf{z}_{\text{DCT}} = \text{zero\_pad}(\mathbf{z}_{\text{DCT}}, D - d)$
$\quad \tilde{\mathbf{x}}_{context} = \text{iDCT}(\mathbf{z}_{\text{DCT}})$
**Return**: $\tilde{\mathbf{x}}_{context}$

---

Due to cropping and quantization operations, the context computation is not invertible anymore. However, we can still go back from the frequency to the local domain. First, we start by multiplying by the normalization matrix $\mathbf{S}$. Afterwards, we pad each channel with zeros, so that the size increases from $d \times d$ to $D \times D$. Lastly, we apply the inverse of the Discrete Cosine Transform (iDCT). We describe this procedure in Algorithm 2. We refer to our top-down hierarchical VAE with a DCT-based context as DCT-VAE.

## 5 EXPERIMENTS

We evaluate DCT-VAE on several commonly used image datasets, namely, MNIST, OMNIGLOT, and CIFAR10. We provide the full set of hyperparameters in Appendix E.2. We designed the experiments to validate the following hypotheses:

1) *Adding the DCT-based context into hierarchical VAE does not harm the performance (as measured by negative loglikelihood)* (sec. 5.1).
2) *DCT-VAE have more active units / higher KL values* (sec. 5.2).
3) *Latent variables of very deep DCT-VAE carry more information about the input data* (sec. 5.3).

In all the experiments, we implement two models: A baseline Very Deep VAE model without any context (denoted by VDVAE) (Child, 2021), and our approach (DCT-VAE) that is a VDVAE with a DCT-based context on top. We keep both architectures almost identical, keeping the same number of channels, resnet blocks, and latent space sizes. In other words, the only difference in the architecture is the presence of the context in DCT-VAE.

---

[1] In this work, we consider the most widely used type-II DCT.

Table 2: The test performance (NLL) on MNIST and OM-NIGLOT datasets and the number of stochastic layers ($L$).

| MODEL | L | MNIST | OMNIGLOT |
|---|---|---|---|
| | | $-\log p(\mathbf{x}) \leq\downarrow$ | |
| **DCT-VAE** (ours) | 8 | **76.62** | **86.11** |
| **Donwsample-VAE** (ours) | 8 | 77.52 | 87.69 |
| Small VDVAE (our implementation) | 8 | 78.27 | 88.14 |
| Attentive VAE (Apostolopoulou et al., 2022) | 15 | 77.63 | 89.50 |
| CR-NVAE (Sinha & Dieng, 2021) | 15 | 76.93 | — |
| OU-VAE (Pervez & Gavves, 2021) | 5 | 81.10 | 96.08 |
| NVAE (Vahdat & Kautz, 2020) | 15 | 78.01 | — |
| BIVA (Maaløe et al., 2019) | 6 | 78.41 | 91.34 |
| LVAE (Sønderby et al., 2016) | 5 | 81.74 | 102.11 |
| IAF-VAE (Kingma et al., 2016) | — | 79.10 | — |

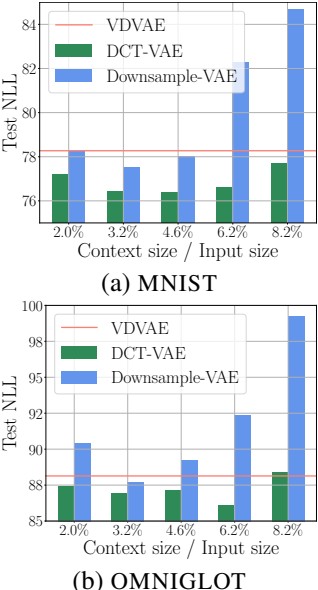

(a) MNIST

(b) OMNIGLOT

Figure 3: NLL results for MNIST and OMIGLOT for different context types and sizes.

## 5.1 IMAGE GENERATION BENCHMARKS

**Binary images** We start with the experiments on binary images: MNIST and OMNIGLOT, for which we use dynamic binarization. In Figure 3, we report the results of an ablation study where we test various context sizes and two contexts: downsampling and DCT. We observe that DCT-VAE (green) outperforms the VDVAE in all the experiments (the orange horizontal line). However, if we choose downsampling as a context instead of the DCT, the performance of the model drops significantly for larger context sizes (blue bars). The reason for that comes from the fact that it becomes harder to fit the prior to the aggregated posterior. Interestingly, it seems there is a *sweet spot* for the context size of the DCT-VAE at around $5\%$. Since DCT always performs better than downsampling, we use it in all the experiments from now on. Comparing DCT-VAE to various best-performing VAEs, it turns out that our approach not only does not harm performance but also achieves state-of-the-art performance on both datasets, see Table 5. Importantly, the introduction of the context gives a significant improvement over the same architecture of the VDVAE.

**Natural Images** We perform experiments on natural images to test the method's performance on a more challenging task. We use the CIFAR10 dataset, which is a common benchmark in VAE literature.

We note that the best-performing VAEs (e.g., VDVAE, NVAE) on this dataset are very large and require substantial computational resources to train which we do not have access to. Instead, we train a small-size VDVAE and provide results of other generative models of comparable sizes in Table 5. We report the complete comparison (including large models) in Appendix D.

Table 3: The test performance (BPD) on the CIFAR10 dataset, the total number of trainable parameters (Size), the number of stochastic layers ($L$).

| MODEL | SIZE | L | BITS/DIM $\leq\downarrow$ |
|---|---|---|---|
| **DCT-VAE (ours)** | 22M | 29 | 3.26 |
| Small VDVAE (our implementation) | 21M | 29 | 3.28 |
| OU-VAE (Pervez & Gavves, 2021) | 10M | 3 | 3.39 |
| Residual flows (Perugachi-Diaz et al., 2021) | 25M | 1 | 3.28 |
| i-DenseNet flows (Perugachi-Diaz et al., 2021) | 25M | 1 | 3.25 |

Table 4: The absolute and the relative number of active units for VAEs and DCT-VAEs evaluated on the test datasets of MNIST, OMNIGLOT, and Cifar10.

| | LATENT SPACE | CONTEXT SIZE | AU↑ (Absolute) | AU↑ (% of latents) | KL↑ (per latent unit) $\times 10e-3$ |
|---|---|---|---|---|---|
| | | | MNIST | | |
| VDVAE | 980 | 0 | 336 | 34.4% | 22.9 (1.4) |
| DCT-VAE | 967 | 36 | **405** | **41.9%** | **25.9** (0.8) |
| | | | OMNIGLOT | | |
| VDVAE | 980 | 0 | 494 | 50.4% | 35.1 (0.8) |
| DCT-VAE | 980 | 49 | **593** | **60.5%** | **36.5** (0.8) |
| | | | CIFAR10 | | |
| VDVAE | 105K | 0 | 7.5K | 7.1% | 47.6 (2.1) |
| DCT-VAE | 105K | 108 | **11.3K** | **10.8%** | **51.6** (2.0) |

We observe that our approach works on par with the generative models that have comparable sizes (OU-VAE, Residual Flows, GLOW), and, most importantly, it has a similar (in fact, slightly better) BPD to our implementation of the VDVAE of a similar size.

## 5.2 POSTERIOR COLLAPSE

In this section, we analyze the latent space of the DCT-VAE and VDVAE trained on different datasets from the posterior collapse point of view. We report the number of active units and KL-divergence on the test dataset in Table 4. We also show the total latent space size and context size.

We observe that the number of active units increases significantly when the context is introduced to the model. Furthermore, this increase is much higher than the size of the context itself, meaning that it helps to increase the latent space utilization in general. However, there are still a lot of unused latent variables. For example, on the CIFAR10 dataset, the proportion of active units increases from 7% to 11%. It means that even though deeper models obtain better NLL, there is still a significant waste of the model's capacity. Similarly to the AU metric, the higher KL-divergence of the DCT-VAE compared to the VDVAE with no context indicates that the DCT-based context helps to *push* more information to other layers. In conclusion, we observe the improved utilization of latent space in terms of both metrics.

## 5.3 DATA INFORMATION IN LATENT VARIABLES

Many of the state-of-the-art models have a lot of stochastic layers (e.g., 45 for CIFAR10 (Child, 2021)). Therefore, it is likely that the information about the $\mathbf{x}$ could be completely disregarded by the latent variables further away from the input. In this section, we explore how much information about the corresponding data points the top latent codes contain. For this purpose, we consider the reconstruction performance and compression. We examine VDVAE and DCT-VAE with 29 stochastic layers trained and tested on the CIFAR10 dataset in both experiments.

### 5.3.1 RECONSTRUCTION CAPABILITIES OF DCT-VAE

We compute Multi-Scale Structural Similarity Index Measure (MSSSIM) (Wang et al., 2003) for the test data and its reconstruction obtained using only part of the latent variables from the variational posterior. That is, for each $m \in \{1, \ldots, L\}$ we obtain a reconstruction $\tilde{\mathbf{x}}^m$ using $m$ latent variables from the variational posterior and by sampling the rest $L-m$ latent variables from the prior, namely:

$$\tilde{\mathbf{x}}^m \sim p_\theta(\cdot|\mathbf{z}_{1:L}) \prod_{l=1}^{L-m} p_\theta(\mathbf{z}_l|\mathbf{z}_{l+1:L}) \prod_{l=L-m+1}^{L} q_\phi(\mathbf{z}_l|\mathbf{z}_{l+1:L}, \mathbf{x}). \tag{8}$$

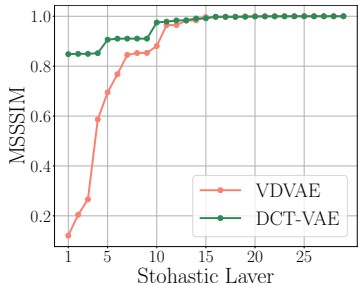 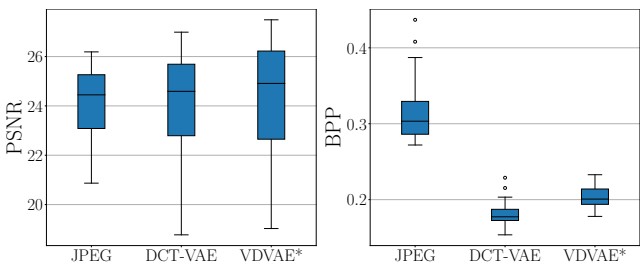

Figure 4: The reconstruction measured by the MSSSIM (↑) on the CIFAR10 test set for a varying number of latent variables sampled from the encoder.

Figure 5: Compression result on KODAK dataset. We use discrete context only to compress images with DCT-VAE. We report the BPP of JPEG and VDVAE that corresponds to the same reconstruction quality.

PSNR = 15.2, MSSSIM = 0.38, BPP* = 0.05          PSNR = 25.1, MSSSIM = 0.84, BPP = 0.19          PSNR = 26.6, MSSSIM = 0.84, BPP = 0.32

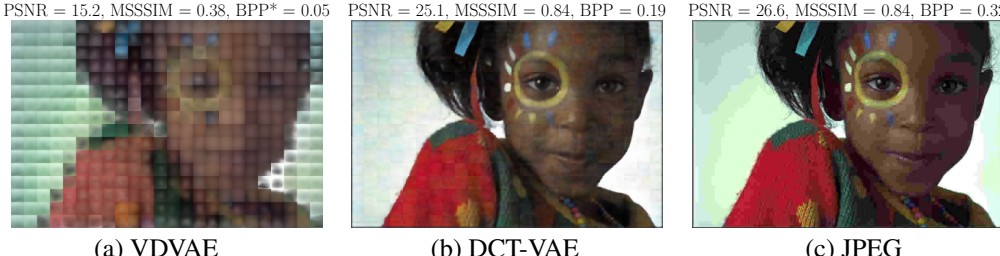

(a) VDVAE                     (b) DCT-VAE                     (c) JPEG

Figure 6: Examples of the decompressed images. We use (a) 2 top latent variables of VDVAE to reconstruct the image, (b) only the context of DCT-VAE, and (c) we choose JPEG compression to have a similar PSNR value to DCT-VAE.

We present the results of this experiment in Figure 4. We observe that in VDVAE the top latent layers carry very little to no information about the real data point $\mathbf{x}$, which continues up to the 5[th] layer from the top. Then, the reconstructions become reasonable (between the 5[th] and the 10[th] layer values of MSSSIM increases from 0.6 to 0.8). In the case of DCT-VAE, using only one layer (i.e., context) gives already reasonable reconstructions (MSSSIM above 0.8).

### 5.3.2 IMAGE COMPRESSION WITH DCT-VAE

To find out how much information about the data is preserved in the top latent variable, we conduct an experiment in which we use the baseline VDVAE and the DCT-VAE pretrained on CIFAR10 for compression. We use the KODAK dataset, which is a standard compression benchmark containing 24 images with resolution $512 \times 768$. Since CIFAR10 images are $32 \times 32$, we independently encode patches of KODAK images. We then reconstruct each patch using only the context latent variable, while the rest of the latent variables are sampled from the prior. We combine these patches to obtain final reconstructions and measure reconstruction error (PSNR). We use JPEG as a baseline.

Results are provided in Figure 5. We select the compression rates that result in comparable PSNR values. We report KL-divergence converted to bits-per-pixel as a theoretical compression rate. All the latent variables (except for the context in DCT-VAE) are continuous. We provide an example of the KODAK image after compression in Figure 6. We also plot examples of the reconstructed images in the Appendix Figure 9. Interestingly, DCT-VAE is capable of obtaining much better BPP than two other baselines while keeping the same PSNR. This indicates the usefulness of context.

## 6 CONCLUSION

In this paper, we discuss the issue of posterior collapse in top-down hierarchical VAEs. We show theoretically and empirically that this problem exists. As a solution, we propose to introduce deterministic, discrete and non-trainable transformations to calculate the top latent variables, e.g., DCT. The resulting model, DCT-VAE, seems to give more robust latent variables that carry more information about data (e.g., the compression experiment).

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
