# OpenReview forum: "Discouraging Posterior Collapse in Hierarchical Variational Autoencoders Using Context"
_ICLR.cc/2024/Conference — ICLR 2024 Conference Withdrawn Submission_

### Official Review · Reviewer_dETJ · 2023-10-17

**Soundness:** 3 good
**Presentation:** 3 good
**Contribution:** 2 fair
**Rating:** 3
**Confidence:** 4

**Summary:**

This paper tackles the problem of inactive units in deep VAEs by introducing in deep VAE architectures a deterministic data-dependent context as a top-level latent variable.
The context is obtained using the Discrete Cosine Transform of the input image, and therefore provides a compressed representation of the image.
Experiments on standard benchmarks show that the model help reducing the inactive units in deep VAEs, and leads to improved log-likelihood performances.

**Strengths:**

The problem of collapsing units in deep VAEs is well known and still an open research question. This paper introduces and demonstrates an interesting direction of investigation that seems to help reduce inactive units.

Knowing that the DCT works well for compressing images, it makes sense to me to use this "domain knowledge" in the model architecture.
To the best of my knowledge, this idea is novel in the context of deep VAE architectures, although Fourier-like transforms were used in many different settings in deep learning.

It is overall a well written paper that is easy to follow.

**Weaknesses:**

As is, the paper needs important changes/improvements in some key areas, that if properly addressed could make this an interesting work to the broader community.

**1. Generalization of experiments to other hierarchical VAE architectures:**

What your experiments show is that for VD-VAEs the context helps (there are still quite a lot of inactive units as also mentioned by the authors), so in a way in this paper you are proposing an improvement of the VD-VAE model. But the claims you make are way more general since they are applied to all hierarchical VAE architectures. This is not correct, unless substantiated by adding the context in at least one other hierarchical VAE architecture (e.g. Ladder VAE, BIVA) and showing a reduction of the active units.

**2. Misleading statements:**

I find the following statements on the premise of the paper quite misleading/untrue
1. Abstract: "There is a consensus that the top-down hierarchical VAEs allow effective learning of deep latent structures and avoid problems like posterior collapse."
2. Section 3.2: "The common claim that the top-down hierarchical VAEs alleviate the problem of the posterior collapse (Maaløe et al., 2019) is not necessarily true"

These statements should be revised, since this is not what is claimed in previous papers. To me the consensus is that top-down hierarchical VAE can still have collapse, but that the problem can be reduced by specific architectures/parameterizations:
1. In the BIVA paper for example, the deterministic top down path and the skip connections are key to avoid collapse. The authors of the BIVA paper say for example in the introduction: "We introduce the Bidirectional-Inference Variational Autoencoder (BIVA), a model formed by a deep hierarchy of stochastic variables that uses skip-connections to enhance the flow of information and avoid inactive units"
2. The BIVA paper claims that the Ladder VAE has active units issues, despite having a top-down hierarchy.



**3. Explanation of why/how the model works:**

The paper would benefit from a more comprehensive explanation of how the model works. Providing a deeper insight into the workings of the proposed deep hierarchical VAE, particularly the role of context, would not only improve the paper but also open up for future research directions. Consider addressing the following:

ROLE OF CONTEXT

   a. *Visualizing DCT-based context:* It's important to visualize the DCT-based context obtained in some of the data samples in the experiments. This can be achieved by applying Algorithms 1 and 2 from section 4.4 on dataset examples with the chosen context size. This visualization would help readers understand how much information is being passed to the model through the context.

   b. *Comparison with a model using only the diffusion prior:* It would be valuable to include an ablation study comparing the proposed model with the same model using a diffusion prior but no context. This comparison could shed light on the specific advantages of the context.

   c. *Extension to other domains:* It would be helpful to discuss how this approach can be extended to other domains. For example, how this model might be adapted for time series data or text.

PRIOR DISTRIBUTION

The paper's discussion of the proposed prior distribution could be more detailed and clear.

   a. *Visualizing the Prior:* It would be helpful to visualize the learned prior distribution in some of the experiments, by computing the iDCT of prior samples.

   b. *Alternative Priors:* What are possible alternative priors? What happens if you use as prior the DCT of mean image over training set?

   c. *Discrete $z_L$:* Explain the necessity of $z_L$ being discrete and of the quantization step. Have you explored using standard continuous priors, and if so, what were the outcomes?

   d. *Unconditional Sample Generation:* It would be helpful to show generated unconditional samples on datasets like MNIST or CIFAR.



**4. Minor comments:**

- "Var" in equation (4) is not defined in the paper
- Section 3.3: $z_{-l}$ not defined
- Section 3.3: In the "Non-identifiability equation" $z$->$z_l$?

**Questions:**

See questions/comments listed in the weaknesses section.

---

### Official Review · Reviewer_aUPB · 2023-10-27

**Soundness:** 2 fair
**Presentation:** 3 good
**Contribution:** 3 good
**Rating:** 5
**Confidence:** 4

**Summary:**

This study presents both theoretical and empirical evidence indicating the persistence of posterior collapse issues within top-down hierarchical VAEs, which diverges from the prevailing consensus. The authors further introduce an approach to discourage posterior collapse by incorporating a contextual element (an input transformation) as a top-level latent variable.

**Strengths:**

- The study regarding the analysis of posterior collapse in hierarchical VAEs is extensive and backed by theoretical and empirical evidence.

- The proposed method to discourage posterior collapse in VDVAE shows improvement empirically.

**Weaknesses:**

- The proposed method to discourage posterior collapse is heuristic with little justification or theoretical basis.

- It is misleading to claim that the proposed method **just** adds a non-trainable transformation over VDVAE. In contrast, the joint objective combined with Eq 13 in the appendix results in a full-fledged latent diffusion model that is jointly trained with the model.


The equation in Section 4.2 needs to be labeled. The joint objective is referred to in the appendix as Eq 7, which is wrong.

**Questions:**

Can you theoretically justify your idea?

---

### Official Review · Reviewer_nW8Z · 2023-10-31

**Soundness:** 2 fair
**Presentation:** 3 good
**Contribution:** 2 fair
**Rating:** 5
**Confidence:** 2

**Summary:**

This paper first shows numerically that posterior collapse can occur even in hierarchical VAEs. Then, the result of Wang et al. (2021) about the equivalence between posterior collapse and non-identifiability for plain VAEs is extended to hierarchical VAEs. Based on these results, this paper proposes DCT-VAE, a hierarchical VAE with a Discrete Cosine Transformer as a context. The proposed method is applied to image datasets and quantitatively evaluated its prediction accuracy and improvement of posterior collapse, and qualitatively evaluated its image generation and reconstruction abilities.

**Strengths:**

- The results in Table 1 and Table 4 support the claim that the hierarchical latent structure is effective in the posterior collapse of VAE using multiple datasets and metrics.
- Figure 6 and Figure 9 show the qualitative improvements over simple VAEs.
- How to make contexts is a matter of design choices. The proposed method uses a DCT-based method, and this choice is justified by an ablation study that compares it with a down-sampling method (Section 5.1).
- The paper is well written, and I was able to understand the contents of the paper easily.

**Weaknesses:**

- This paper claims hierarchical VAE resolved posterior collapse (e.g., in Abstract and Section 1). However, I have a question about whether the proposed method solves the posterior collapse problem of hierarchical VAE for the first time. Wang et al. (2021) uses hierarchical VAEs in their experiments. Kinoshita et al. (2023) also improved the posterior collapse of hierarchical VAE theoretically and practically by imposing inverse Lipschitz-ness on the decoder. Other hierarchical VAEs proposed in existing studies mentioned in this paper may alleviate the problem, although they do not address the posterior collapse problem.
- In the image compression task, the proposed method does not improve over existing methods (Table 3, Table 5).
- Due to the computational resource problems, the proposed method is not validated on large models (~100M parameters.) Therefore, it is not known whether the proposed method is effective in more practical cases.


(Kinoshita et al. 2023): https://proceedings.mlr.press/v202/kinoshita23a.html

**Questions:**

* How hyperparameters of the proposed method in Section E.2 are determined?
* How hyperparameters of baseline methods in Table 2 are determined?

【Minor Comments】
* P.4, Section 3.3: $\boldsymbol{z}_{-l}$ is undefined.
* P.5, Section 4.2: Clarify what $\gamma$ in $\mathcal{L}(\gamma, \mathbf{z}_l)$ means.
* P.7, Section 5.1: It is better to clarify that Table 5 is in the Appendix.
* P.13, Section D: in n Table 5 -> in Table 5

**Details Of Ethics Concerns:**

N.A.

---

### Official Review · Reviewer_KzhA · 2023-10-31

**Soundness:** 2 fair
**Presentation:** 1 poor
**Contribution:** 3 good
**Rating:** 5
**Confidence:** 2

**Summary:**

The paper presents an VAE variant to address posterior collapse issue in the existing works. The paper starts with demonstrating the posterior collapse in top-down hierarchical VAEs by adopting active units metrics and plotting kl divergence. With the empirical evidence demonstrated, the authors then described the connection between posterior collapse and non-identifiability of latent representations. With all of the above works, the proposed method is presented where context information is encoded using classic non-invertable DCT transformation, such that the latent collapse could be voided. Experiments show that using proposed method (DCT-VAE), the model is able to use more latent expressions than it was without DCT.

**Strengths:**

1. Leveraging classic signal processing method to compress input as context is something useful in practice. It is great that VAE like model to adapt it and shows positive performance gain.
2. The paper provides sufficient motivation on the lack usage of latent representations.

**Weaknesses:**

1. The paper is poorly written and structured. Sections lack connections. Figures lack description. E.g. what is d1 and d2 in Figure 2? Where is the diffusion-based model described? If it is important to highlight in the paper, which equation shows the usage of such model? What is the purpose of describing Section 3.3 if its conclusion is not used in the following section? All of the above confusions makes me hard to be convinced with the contribution of this work unfortunately.
2. Empirical results show the proposed method is not such significant in terms of addressing collapse problem. For example, for MNIST (Table 4), the AU is improved from 34% to 49% but still with 50% gap to the full usage of latent representations. for CIFAR10, the proposed method only makes 10.8% latent usage. Is it really a good result?

Indeed, this might be a good work, but the writing quality really hurts my reading experience.

**Questions:**

My questions are listed above in weakness.